# The Role of Nucleocapsid Protein (NP) in the Immunology of Crimean–Congo Hemorrhagic Fever Virus (CCHFV)

**DOI:** 10.3390/v16101547

**Published:** 2024-09-30

**Authors:** Aysegul Pirincal, Mehmet Z. Doymaz

**Affiliations:** Department of Medical Microbiology, School of Medicine and Beykoz Institute of Life Sciences and Biotechnology, Bezmialem Vakıf University, Istanbul 34093, Türkiye; aysegul.pirincal@bezmialem.edu.tr

**Keywords:** *orthonairovirus*, CCHFV, nucleocapsid protein, immunity, vaccine

## Abstract

Crimean–Congo hemorrhagic fever virus (CCHFV) is an *orthonairovirus* from the *Bunyavirales* order that is widely distributed geographically and causes severe or fatal infections in humans. The viral genome consists of three segmented negative-sense RNA molecules. The CCHFV nucleocapsid protein (CCHFV NP) is encoded by the smallest segment of the virus. CCHFV NP, the primary function of which is the encapsidation of viral RNA molecules, plays a critical role in various mechanisms important for viral replication and pathogenesis. This review is an attempt to revisit the literature available on the highly immunogenic and highly conserved CCHFV NP, summarizing the multifunctional roles of this protein in the immunology of CCHFV. Specifically, the review addresses the impact of CCHFV NP on innate, humoral, and cellular immune responses, epitopes recognized by B and T cells that limit viral spread, and its role as a target for diagnostic tests and for vaccine design. Based on the extensive information generated by many research groups, it could be stated that NP constitutes a significant and critical player in the immunology of CCHFV.

## 1. Introduction

The Bunyavirales order, as defined by the International Committee on Taxonomy of Viruses (ICTV), encompasses a group of enveloped viruses with segmented, single-stranded RNA genome that exhibits negative or ambisense polarity. This order includes various pathogens capable of causing severe infectious diseases in humans, animals, and plants [1,2]. Except for Arenaviridae, other bunyaviruses that are pathogenic to humans, including *Nairoviridae*, *Hantaviridae*, *Peribunyaviridae*, and *Phenuiviridae*, are characterized by their genome organization, which encodes the RNA-dependent RNA polymerase (RdRp) enzyme on the large (L) segment, surface glycoproteins on the medium (M) segment, and nucleocapsid (NP) protein on the small (S) segment, according to their size [3]. Genomic RNA molecules, together with the NP, form NP–RNA complexes, which are packaged as viral ribonucleoproteins (vRNPs). The virus’s RNA-dependent RNA polymerase enzyme is also transported in association with vRNPs [4]. Within the order Bunyavirales, the Nairoviridae family is classified into seven serogroups based on antibody cross-reactivity [5]. 

CCHFV is a tick-borne orthonairovirus capable of causing hemorrhagic fever (CCHF) in humans, with mortality rates reaching up to 40% and a recent increase in incidence [6,7]. *Hyalomma marginatum* ticks are natural hosts and major reservoirs of CCHFV and play a significant role in the transmission of the disease [8,9]. H. marginatum exhibits a broad geographical distribution encompassing Southern Europe, North Africa, Anatolia, the Caucasus, and the former Soviet republics. Although these ticks can be encountered on animals from February to December, adults are predominantly active in blood-feeding from March to August, whereas larvae and nymphs are active from June to November [10]. Studies have shown that a single tick-specific amino acid variant identified in the viral glycoprotein region can significantly reduce membrane fusion activity in human cells, potentially leading to impaired infectivity in ticks [11]. The primary target cells of CCHFV are vascular endothelial cells and hepatocytes [12]. The clinical symptoms of CCHF include prolonged high fever accompanied by bleeding and vascular permeability indicative of viral involvement. The clinical progression of CCHF is delineated into four distinct stages. Initially, the infection and incubation period occur, with the incubation phase typically lasting 1–7 days following exposure. Subsequently, individuals may enter a pre-hemorrhagic stage, marked by nonspecific symptoms including fever, malaise, myalgia, and nausea. This pre-hemorrhagic phase can rapidly evolve into the hemorrhagic stage within the first week post-infection. The hemorrhagic stage is characterized by severe manifestations, such as uncontrolled bleeding, hepatic damage, intense inflammatory immune responses, and, in extreme cases, may culminate in death. Recovery in surviving patients generally commences 10–20 days after infection, during which clinical symptoms gradually normalize [13]. 

The global geographic distribution of CCHF notably includes over 30 countries, primarily in Africa, Asia, Eastern Europe, and the Middle East [14]. The disease is most frequently reported from Türkiye. However, the presence of the infection is recognized in numerous countries, ranging from Southern Europe, including Spain, to Japan [15,16,17]. The World Health Organization (WHO) “Priority Diseases” report [18] highlights that CCHF ranks second after COVID-19, emphasizing the importance of research and development in this field.

Over the past two decades, intensive research on CCHFV was conducted and excellent reviews have been published on various aspects including, as follows: the molecular structure, genetic diversity, and pathogenesis of CCHFV [4,19,20]; tick–host interactions [21]; animal models and seroepidemiologic studies [20,22,23]; immunological responses [8,24,25]; diagnostic approaches [26]; and antivirals–vaccine developments [6,27,28,29]. This review, however, aims to readdress the existing literature on the highly immunogenic and highly conserved CCHFV NP. In this context, a comprehensive framework of the NP protein is provided by discussing its viral biology and structural characteristics, functional differences among other viral NPs, roles in viral replication, interactions with cellular proteins and factors, and their impacts on viral pathogenesis. The multifunctional roles of NP in CCHFV immunology, which form the main focus of our review, are examined with respect to its effects on innate, humoral, and cellular immune responses, immunological epitopes that limit viral spread, and its functional use in vaccine development and diagnostic tests, emphasizing why the NP protein is an important immunological target. 

## 2. CCHFV NP in Viral Biology

The CCHFV nucleocapsid protein (CCHFV NP) is the most abundant protein in the virion. CCHFV NP has a molecular weight of approximately 54 kDa and is composed of 482 amino acid residues. The protein consists of a stalk domain (residues 180–300), and a head domain covered of N-terminal 180 amino acids and C-terminal 182 amino acids (residues 1–180 and 300–482). CCHFV NP is predominantly localized in the perinuclear region of infected cells [4]. In addition to the structural NP, the CCHFV S segment encodes an NSs protein through ambisense reading in the opposite direction to NP. The NSs protein, which is a non-structural protein of CCHFV, exhibits localization in the mitochondria. Its known primary function is to trigger apoptosis by disrupting mitochondrial membrane potential [30] (Figure 1). The infection of host cells by CCHFV and the virus replication cycle can be generally outlined as follows: (1) attachment of virions to cell-surface receptors; (2) entry into the cell via clathrin-mediated endocytosis and membrane fusion; (3) primary transcription; (4) translation of viral proteins and post-translational modifications; (5) replication of viral RNAs; and (6) assembly and release by budding into the Golgi, followed by exocytosis. The replication of genomic RNPs involves the replication and encapsidation of negative-sense viral RNA (vRNA) and an intermediate molecule, antigenomic RNA (cRNA). In this context, the presence of the newly synthesized NP protein, alongside the L protein, is essential for CCHFV replication and the viral life cycle [4].

A common feature among all bunyaviruses for NPs is the significant role they play in genome encapsidation and vRNP formation; however, there are significant differences in structure and function among their NPs [31]. 

Unlike the CCHFV NP, the positively charged pocket was observed to be closed in the head domain of the Erve virus (ERVEV) NP, suggesting a difference in the DNA-specific endonuclease activity of ERVEV NP [32]. Although the head domain of CCHFV NP shares high structural similarities with the N-terminal domain of Lassa fever virus NP, it has been shown that the C-terminal domain possesses dsRNA-specific 3′ to 5′ exoribonuclease activity, which is essential for immune suppression [33]. In the structure of CCHFV NP, the head-stalk orientation presents the straightest architecture. However, significant conformational differences appear in the orientation between the head and stalk domains of other viral NPs. For instance, compared to CCHFV NP, the stalk domains of Kupe virus (KUPV) and ERVEV NPs are rotated approximately 30°, while the stalk domain of Hazara virus (HAZV NP) is rotated by more than 60°. Additionally, the stalk domain of KUPV NP adopts a different conformation, rotated approximately 40° within an asymmetric unit, which is thought to contribute to high structural flexibility during RNA binding or oligomerization. Experiments using a small RNA probe showed that KUPV NP has a higher binding affinity compared to CCHFV NP’s binding [32]. The caspase 3 cleavage site serves as an example of how structural differences between CCHFV NP and other viral NPs lead to functional variations. Despite the presence of a similar loop between helices α11 and α12 at the distal end of the stalk domain in both KUPV NP and CCHFV NP, the P1 residue in the potential caspase 3 cleavage site of KUPV NP is Glu rather than Asp. This substitution suggests that KUPV NP might not be susceptible to cleavage by caspase 3. Likewise, DUGV NP is also resistant to recognition and hydrolysis by caspase 3. These findings imply that the innate immune response observed in CCHFV infections, which targets the NP’s stalk domain, may not be effective against other nairovirus serogroups [32].

Although their roles in CCHFV replication are not yet fully understood, the CCHFV NP possesses two distinct RNA-binding regions for single-stranded and double-stranded RNA molecules [34,35]. In the absence of RNA, NP exists in a monomeric form and binds weakly to RNA. It has been shown that oligomerization induces a conformational change in NP, allowing it to bind more strongly to RNA. Thus, oligomerization appears to regulate NP function [36]. While the chemical interactions between the RNA molecule and NP are not fully understood, mutations in CCHFV NP residues (K132, Q300, K411), which are thought to play a role in RNA binding, have been shown to block the transcription and replication of CCHFV minigenomes [37]. Additionally, in the presence of the cellular translation initiation factor eIF4G, CCHFV NP binds with higher affinity to the 5′-UTR sequences of viral mRNAs, thereby promoting the initiation of translation [38]. The low binding affinity to the 5′ cap of CCHFV NP in its monomeric form suggests that it may also operate through a cap-independent mechanism in translation processes. Thus, CCHFV develops an NP-mediated translation strategy for facilitating the rapid synthesis of viral proteins. This strategy, which is crucial for viral pathogenesis, also holds potential as a therapeutic target in combating the pathogenesis [38]. 

Viruses utilize the host cell’s biosynthesis machinery to replicate. As a result, during the replication cycle, from the initial adsorption phase of cell entry to the release of new viral particles, it is inevitable that proteins encoded by the viral genome interact with numerous host cell proteins. This year, a significant advance in understanding the biology and pathogenesis of CCHFV was made with the identification that the virus utilizes the low-density lipoprotein receptor (LDLR) for cell entry and employs apolipoprotein E (apoE), a natural LDLR ligand, to facilitate cellular entry. It has been indicated that, following the high-affinity binding of the viral surface glycoprotein Gc to the cellular LDL receptor, the virus is internalized into the host cell via endocytosis [39,40,41]. From the perspective of virus–host interactions, the identification of cellular proteins associated with the NP, which is abundantly synthesized from the early stages of viral infection, is of great importance for understanding the viral life cycle and for developing new approaches to combat CCHFV. Studies have elucidated several cellular metabolic pathways affecting viral replication, including those involving CCHFV NP and some associated cellular proteins.

Heat shock protein 70 (HSP70)**,** a cellular chaperone, has been shown to interact with both CCHFV NP and HAZV NP. Furthermore, the use of inhibitors that block HSP70 function has been demonstrated to significantly reduce virus titers [42]. Thus, leveraging this interaction, which is crucial and highly significant for nairovirus replication, as a strong target in the development of therapies will be extremely important in combating these viruses. It has been determined that CCHFV NP interacts with actin filaments, and that actin filaments play a role in targeting viral NP to perinuclear regions. Additionally, the presence of agents that disrupt actin filaments has been reported to reduce viral assembly by up to 97% [43]. Regions involved in the homo-oligomerization of CCHFV NP, binding with actin filaments, and the formation of NP-L protein complexes with viral polymerase, have been identified. The involvement of CCHFV NP in viral replication has been demonstrated through its interaction with the N-terminal regions of the L protein and co-localization with the L protein in transfected cells. These findings are particularly valuable for guiding the development of potent inhibitors that could block NP–L complex formation, and, consequently, CCHFV replication [44,45].

## 3. CCHFV NP in Innate Immunity

The relationship between CCHFV NP and innate immunity is among the least explored areas within the broader context of NP’s interactions with immune responses. This characterization can also be applied to CCHFV’s interactions with natural immunity in general. As extensively discussed in a previous review [46], it should also be emphasized that the most compelling evidence of the paramount importance of innate immunity in CCHFV infection has been obtained from studies using CCHFV-interferon alpha/beta-deficient mouse models. Since the initiation of studies on CCHFV in mouse models, it has been observed that IFN–alpha/beta-competent mice are refractory to infection. Researchers have addressed this issue by either temporarily neutralizing the IFN–alpha/beta receptor with monoclonal antibodies or by permanently eliminating the receptor through genetic modifications. Therefore, the CCHFV-innate immunity relationship should be evaluated in the light of these findings. For example, in a model developed by Hawman and colleagues using type I interferon-deficient mice, infection with the Hoti strain led to a disease characterized by prominent clinical symptoms lasting several days [47]. IFNAR^−/−^ mice infected with CCHFV also exhibited clinical symptoms similar to those of Crimean–Congo hemorrhagic fever, while wild-type mice remained asymptomatic, highlighting the importance of type I IFNs in preventing CCHFV infection [48]. Similarly, Bente and colleagues confirmed the critical role of IFNs in combating CCHFV by demonstrating that STAT-1 is a crucial component of IFN-signaling pathways, using the STAT-1 knockout mouse model [49]. 

With the entry of CCHFV into the cell, viral components known as pathogen associated molecular patterns (PAMPs) are recognized by pathogen recognition receptors (PRRs), which serve as antigen recognition receptors, primarily on immune cells such as monocytes, macrophages, and dendritic cells. This recognition triggers the activation of innate immune-signaling mechanisms, leading to the production of various pro-inflammatory cytokines, chemokines, and antiviral responses. During viral infection, immune cells secrete cytokines such as TNF-α, IL-1, IL-6, IL-8, IL-12, IFN-γ, MCP-1, and MIP-1b. Studies have reported that levels of IL-1β, IL-5, IL-6, IL-8, IL-9, IL-10, IL-15, IP-10, MCP-1, TNF-α, and RANTES significantly differ between fatal and non-fatal cases. Notably, differences observed in IP-10 and MCP-1 levels between surviving patients with severe and mild cases have been significantly associated with viral load, disease severity, and outcome, suggesting that these cytokines may serve as biomarkers [50]. Additionally, in the early stages of the disease (first 5 days), levels of IL-6 (92%) and IL-8 (92%) were found to be significantly higher in fatal cases compared to survivors. Thus, elevated levels of IL-6 and IL-8 are considered potential prognostic markers [51]. Additionally, in infected mice, the activation of mitochondrial antiviral signaling protein (MAVS) and the TNF-α receptor have both been implicated in contributing to CCHFV pathogenesis, leading to severe liver damage [52]. 

The activation of innate immunity is crucial for the rapid and effective containment of viral infection and the subsequent initiation of adaptive immune responses. Like many other negative-sense RNA viruses, CCHFV is detected by distinct receptors of the innate immune system: RIG-I-like receptors (RLRs) and Toll-like receptors (TLRs) [52,53].

Toll-like receptors (TLRs) are type I transmembrane proteins consisting of a transmembrane domain responsible for signal transduction. They are classified into two groups based on their intracellular localization and the ligands they target. The group including TLR3, TLR7, TLR8, and TLR9 is found in intracellular vesicles such as lysosomes, endosomes, and the endoplasmic reticulum. These receptors recognize single- or double-stranded viral nucleic acids from various RNA viruses, including CCHFV. It has been proposed that the recognition of viral pathogens by TLRs triggers a series of reactions that lead to the phosphorylation of the IRF-3/7 transcriptional activator, thereby initiating the expression of IFN-β. TLR polymorphisms have been widely studied in the context of infectious diseases [54]. There are publications suggesting that certain polymorphisms in the TLR gene family may be significant in determining susceptibility to CCHF and the clinical course of the disease in patients in Türkiye [55,56]. However, these studies need further validation. Within the RLR family, RIG-I, an RNA helicase enzyme, is a cytoplasmic receptor whose synthesis is induced by retinoic acid. It has been proposed that RIG-I is particularly sensitive to the 5′-triphosphate (5′-PPP) group at the 5′ end of ssRNA generated during CCHFV replication and that it initiates antiviral mechanisms against the virus [57]. However, there are also publications indicating that CCHFV particles prevent RIG-I from binding to CCHFV RNA by facilitating the removal of the 5′-triphosphate group from genomic RNA after transcription [58]. 

During the innate immune response, the induction of the interferon (IFN) response and the subsequent emergence of a range of IFN-dependent responses are critical events. However, it has been noted that CCHFV delays the nuclear translocation of IRF-3 (interferon regulatory factor 3), a transcriptional activator, thereby delaying the IFN response and facilitating rapid viral dissemination within the host cell [59]. Another study specifically investigated the NP of CCHFV in this context. It was reported that while the NP of the CCHFV Hoti strain suppresses the activity of the IFN-β promoter, NPs from the IbAr10200 and AP92 strains do not have such an effect on the IFN-β promoter [60].

The pathogenesis and virulence of CCHFV is a dynamic process involving a multitude of factors, some related to the host cell and others to the virus itself. Despite the protection provided by innate immune responses from host cells, CCHFV can evade immune surveillance through the development of various mechanisms in viral pathogenesis. The OTU domain, a cysteine protease located in the N-terminal region of the CCHFV L protein, has been identified as a viral antagonist against innate immunity. The OTU domain is crucial for viral pathogenesis due to its de-ISGylation and deubiquitinase activities [61,62]. However, it is well-known that the non-structural proteins of viruses often play roles as viral antagonists. However, such a function has not yet been reported for the NSs protein, which is a non-structural protein encoded by the S segment of the CCHFV [63]. In a study of mouse-adapted CCHFV, a sequence analysis of the viral segments suggested that the mutation identified in NSs was considered to have the capability to antagonize innate immune responses in mice [46]. 

In response to viral infections, certain antiviral molecules synthesized through interferons to protect the host cell can exhibit effects that either slow down or completely inhibit viral replication. Mx proteins, antiviral molecules whose synthesis is induced by interferons (ISGs) and affect different stages of viral replication and translation, are a group of proteins first identified in mice in the context of defense against viral infections [64]. Studies have reported that MxA proteins inhibit the replication of many RNA viruses [65]. It has been demonstrated that MxA protein exhibits antiviral activity by blocking the functions of the viral protein through its localization with CCHFV NP in the perinuclear region of infected cells. Additionally, it has been emphasized that the MxA-645 position is critical in this interaction [66].

CCHFV regulates apoptosis through mechanisms it has developed for viral fitness. In this context, the roles of the NP and non-structural NSs protein, both encoded by the S segment of the virus, in apoptosis are worth examining within the framework of viral infection. The anti-apoptotic CCHFV NP can inhibit caspase 3/9 activation, induction of apoptosis caused by Bax, and the release of cytochrome C from the mitochondria [67]. Additionally, the highly conserved _266_DEVD_269_ motif within the stalk domain of the NP is the caspase 3 cleavage site and is crucial for the viral life cycle. Studies suggest that, unlike its oligomeric form, the monomeric form of the NP enables the DEVD motif, which is a target for caspase 3 enzymes, to be exposed conformationally. This exposure could potentially play a role in the regulation of apoptosis [36]. The CCHFV NSs protein, which exhibits an apoptotic role with localization in the mitochondria, has been reported to disrupt mitochondrial membrane potential, activate caspase 3/7, and cleave poly(ADP-ribose) polymerase [30,68]. However, it has also been suggested that CCHFV inhibits apoptosis during the early stages of infection but exerts a pro-apoptotic effect in the later stages [67].

## 4. CCHFV NP in Humoral Immunity

In addition to innate immunity, which is the initial protective response against CCHFV infection, a robust adaptive immune response is also necessary for effective control of the infection. Following antigenic stimulation, the production of virus-specific antibodies by B cells represents a crucial phase in the humoral immune response against viral infection. Antibodies can prevent infection through mechanisms such as virus neutralization, opsonization, and virus inactivation, some of which are well-characterized while others are not fully understood. Additionally, antibodies contribute to the host defense by facilitating the destruction of infected cells.

Anti-CCHFV IgM antibodies are detected starting 4–5 days after the onset of symptoms and are often observed 7–8 days following the onset of symptoms. However, these antibodies begin to decline from the third week and become undetectable 3–5 months after the onset of the disease. On the other hand, anti-CCHFV IgG antibodies are detected at around days 7–9. IgG titers peak 2–5 months after the onset of the disease and have been observed to remain at detectable levels for up to 3 years [69,70]. 

For a large number of negative-sense RNA viruses, the NP is considered a primary major antigen in viral infections. Various vaccines targeting NP exclusively have been identified for influenza virus, Ebola virus, Lassa virus, and Rift Valley fever virus [71,72,73,74]. According to the results of the studies, these vaccines have been found to elicit strong antibody responses against NP. Additionally, challenge studies have observed that these vaccines induce protective immunity. Another example of humoral immunity against NP is hantavirus infection. During the acute phase of hantavirus infection, IgM and IgG antibodies that react with hantaviral NP are detectable, while antibodies against Gc and Gn appear later, as the disease progresses [75]. 

The capacity of CCHFV NP to induce humoral immunity is among the earliest studies conducted on this virus. In their research, Saijo et al. highlighted the role of NP in humoral immunity by examining the IgM and IgG responses that develop against recombinant NP. The authors associated CCHFV NP with IgM responses involved in the clearance of viremia [76]. In patient sera, it has been reported that the high-titer anti-NP IgG response is particularly influenced by the 201–306 amino acid residues (NP^201–306^) located in the central region of the protein, which is highly conserved among various isolates [77]. 

Another CCHFV study, where severe anti-NP IgG responses were detected, is a study using the cynomolgus macaque model [78]. The role of a DNA vaccine encoding NP and GPC (glycoprotein) antigens in protecting against CCHFV infection has been investigated using a non-human primate (NHP) model. The vaccine has been suggested to be safe and effective in preventing the disease symptoms caused by CCHFV. In this regard, the study represents the first vaccine identified as effective in NHPs [78]. 

Our laboratory has been investigating the humoral immune responses elicited against CCHFV NP in detail for some time. Following immunizations in experimental animals and in patients who have recovered from CCHFV, high-titer NP-specific antibodies are detected when examined for the humoral immune response (Figure 2). Based on these data, NP has been assessed as a highly functional target in the investigation of CCHFV immunology [79]. In another of our studies addressing the immunological similarities among NPs within the nairovirus family, cross-reactivities in humoral immunity were scrutinized. Strong cross-reactivity was observed in the humoral immune response against the NP of Hazara virus, which shares a 60% amino acid sequence similarity with CCHFV NP across different hosts [80]. 

Neutralizing antibodies (NAbs) are generally considered to be those that prevent viral adsorption or the fusion of the viral membrane with the endosomal membrane. Antibodies against internal viral proteins, such as NP, may not possess neutralizing activity on their own. Studies have reported that antibodies targeting CCHFV NP generally lack neutralizing activity [81,82]. However, non-neutralizing antibodies are reported to provide protection through Fc-mediated mechanisms such as antibody-dependent cellular cytotoxicity (ADCC), antibody-dependent cellular phagocytosis (ADCP), and antibody-dependent complement deposition (ADCD) [83]. 

In a study investigating the kinetics of the immunological response involved in the protective efficacy of a vaccine, where a viral replicating particle (VRP) was used, it was noticed that VRP-RNA did not elicit innate immune responses and did not induce significant changes in cytokine levels. However, the VRP vaccine was found to primarily elicit cellular and humoral immune responses by targeting the NP. On the 7th day post-vaccination, NP-specific IgM and IgG antibodies were detected. The study characterized the NP-specific IgG antibodies induced by the vaccine and investigated their role in protection by examining their IgG subclass profiles. Unlike IgG3, NP-specific IgG1, IgG2b, and IgG2c isotypes were identified. These data suggest that NP-specific antibodies may possess Fc-mediated antibody function differentially. Furthermore, it was noted that the non-neutralizing antibodies induced by the vaccine exhibited ADCD and ADCP activities. In the light of the obtained data, it is suggested that NP antibodies induced by the CCHFV-NP-based VRP vaccine play a role in protection through Fc-mediated effector functions [84]. 

The lack of an FDA-approved vaccine and specific antiviral treatment for CCHFV raises the prospect of utilizing CCHFV-immunoglobulins in new-generation therapeutic approaches such as immunotherapy. Thus, antibody-based treatments derived from the plasma of individuals who have recovered from the disease have long been used in the treatment of people infected with CCHFV [85]. Similarly, the protective efficacy of the GP38-specific mAb-13G8 antibody has been investigated. According to the results, the non-neutralizing mAb-13G8 antibody provided over 90% protection in mice deficient in type I interferon [86]. Additionally, the potential of CCHFV NP as a target for antibody-based therapeutics has also been investigated. Studies using the NP-specific mAb-9D5 antibody revealed that early viral spread was restricted primarily to the liver and spleen, and that lesion development was delayed, thereby protecting mice from viral infection. In the characterization of the immunological response contributing to the observed protection, FcR^−/−^ and C3^−/−^ mice were used, and it was determined that protection was provided independently of Fc receptor and complement activity. Additionally, it was noted that the antibody could bind to the NP of various CCHFV strains across a broad spectrum and that it exhibited strong cross-protection against the heterologous CCHFV Afg09-2990 strain. These data suggest that CCHFV NP could be a valid target for antibody-based therapeutics [87]. 

A testament to the value of NP as a primary antigen with which to assess humoral immune responses is the fact that NP has been the most-utilized diagnostic marker in many studies. The presence of specific anti-NP antibodies is commonly used not only in research on humoral immunity but also in serological studies and in the diagnosis of the disease. The serological cross-reactivity of two recombinant CCHFV NP antigens derived from genetically and geographically distinct CCHFV isolates (South African and Greek (AP92) isolates), as identified by Rangunwala and colleagues, indicates that these antigens may be applicable for global diagnostic and epidemiological purposes [88]. Similar studies are detailed in the section titled ‘CCHFV NP Immunology and Diagnostic Tests’.

## 5. CCHFV NP in Cellular Immunity

Although subunit vaccines may trigger neutralizing antibody responses, their failure to provide protection against the lethal infection indicates that antibody responses alone may not be sufficient to control CCHFV infection [89]. Similarly subscribing protective roles to both humoral and cellular immune responses, a different study described a model where modified vaccinia Ankara virus vector expressing CCHFV glycoprotein antigens (MVA-GP) were effective [90]. Interestingly, in these experiments, CCHFV NP was reported to be an ineffective antigen despite its immunogenic potential [91]. In another vaccine study providing complete protection, it was emphasized that neutralizing antibodies alone are insufficient for protection against the CCHFV challenge, and that a Th1 response may be crucial for effective protection [92]. Similarly, it has been demonstrated that following CCHFV infection, T cells are strongly activated, proliferated, and differentiated to produce Th1-type cytokines. These findings were described as indications that T cells may be essential for the survival of mice during acute CCHFV infection and that IFN-γ might be a key antiviral cytokine at this stage [93]. 

Nucleocapsid proteins are recognized as predominant antigens in many RNA viruses. The role of NP in T cell responses has been extensively studied in other negative-strand viruses such as influenza, measles, Ebola, Lassa, and Hantaan virus. It has been demonstrated that NP-specific T cell responses are valuable in protecting against infections. Consequently, various NP-targeted vaccines have been developed using databases such as the Immuno Epitope Database and Analysis Resource (IEDB) to identify human immunodominant CD8^+^ or CD4^+^ T cell epitopes [94]. 

The role of T lymphocytes in determining the immune response to CCHFV infection has also been demonstrated by the detection of memory T cell responses, particularly to CCHFV NP peptides, in patients who survived the disease even 13 years after infection. The study reported evidence of long-lasting memory CD8^+^ T cell responses specific to Puumala virus, another member of the bunyaviruses, persisting for up to 15 years post-infection [95], highlighting that effective long-term protection against the infection can be mediated by T cells. The identification of a greater number of T cell epitopes of NP compared to those of glycoproteins further underscores the importance of NP in memory responses [96]. 

The role of CCHFV NP as a candidate antigen in challenge experiments has been addressed in many studies. As mentioned earlier, Dowall and colleagues compared GPs and NP in their protective efficacy in a mouse model utilizing MVA-GP and MVA-NP. Although the MVA-NP vaccine candidate elicited strong humoral and cellular immune responses in mice, it was indicated as not protective in a challenge with CCHFV [90,91]; CCHFV glycoproteins induced antigen-specific responses and provided protection against the challenge [90]. A recent study reported somewhat contradictory results that the CCHFV NP-VRP vaccine generated protective NP specific T-cell and antibody responses with Fc-mediated functionality [84]. The report indicated that glycoprotein-specific responses were not detected; therefore, the authors concluded that NP is the key immunogen in the protection against a CCHFV challenge [84]. It should be noted that some of the variations observed in studies might originate from the fact that all experiments are not performed uniformly. The carrier vector systems for antigens differ. The amount of the challenging dose also varies; some use lower challenge doses, while others utilize higher ones. Another variation could be introduced by the challenging viral strain, which might differ in its virulence. Additional sources of variation in these unnatural infection models might be the models themselves, where transgenic or experimentally immunosuppressed small animals or NHPs, among others, are used.

Recently, we have been studying the adaptive immune responses against CCHFV NP in surviving patients and in experimental animal models. In our investigation of cellular immune responses to NP, we addressed the nature of T cell responses both in vivo and in vitro using techniques such as delayed-type hypersensitivity (DTH) responses and lymphocyte proliferation assays (LPA). According to our findings, CCHFV NP exhibits strong stimulatory properties for CD4^+^ cells. Specifically, it induces significant increases in levels of IL-17, IL-2, IFN-γ, and IL-4, thereby eliciting various types of cellular immune responses in different species [79]. In another study, where we compared cellular immune responses against CCHFV NP and HAZV NP, immune lymphocytes obtained from mice immunized with recombinant CCHFV NP and HAZV NP were stimulated in vivo or in vitro with CCHFV-rNP or HAZV-rNP. DTH and LPA assays were conducted to analyze these responses. According to the results obtained, strong DTH and LPA responses were detected against both homologous and heterologous stimulating antigens in all test groups. In the study, where cytokine levels were also assessed at the mRNA level, a notable increase in IL-2, IFN-γ, and IL-17A responses was observed [97]. These studies reiterate the potential of NP to elicit a comprehensive cellular immunity.

## 6. CCHFV NP and Protective Immunological Epitopes

Studies have demonstrated that CCHFV NP is a pivotal viral antigen in both humoral and cellular immune responses, serving as one of the fundamental antigenic components of the virus. Therefore, the identification of both B-cell epitopes, recognized by antibodies, and T-cell epitopes on CCHFV NP is crucial. Such identification not only provides a significant molecular foundation for understanding the structure and function of the viral NP but is also highly relevant for vaccine development efforts. Studies aimed at this objective have identified several functional regions on the CCHFV NP. Specifically, the B-cell epitopes of the NP from the CCHFV-YL04057 strain were investigated by Wei et al. using a series of truncated recombinant NPs expressed in *E. coli*. The results, obtained through Western blotting with a polyclonal mouse serum and two monoclonal antibodies (14B7 and 43E5), indicated that the 235–305 region of the NP is highly antigenic [98]. In a parallel study using similar methods, multiple minimal B-cell epitopes were identified in the N- and C-terminal regions corresponding to the head domain of the NP. This finding has contributed to the development of CCHFV multi-epitope peptide vaccines [99]. In a separate study analyzing common antibody epitopes in the NPs of CCHFV, Nairobi sheep disease virus, and Dugbe virus, which all belong to the Orthonairovirus genus, as does CCHFV, noted that CCHFV-NP-specific antibodies exhibit limited cross-reactivity with other nairovirus NPs [100]. 

An earlier study utilized a baculovirus vector to express the full-length AP92-NP from the Greek CCHFV virus isolate, and peptides derived from NP. The approach was reported to be significantly more effective in detecting CCHFV-specific antibodies in positive human sera from Europe, the Middle East, Asia, and West Africa. However, the study noticed weak cross-reactivity with CCHFV and Hazara NPs [101]. 

In predicting the B cell epitopes of CCHFV NP recognized by antibodies, in silico approaches using molecular docking and immunoinformatic tools are also employed [102]. In a study employing truncated recombinant NP and epitope prediction software, the region between amino acids 123 and 396 was highlighted as a highly immunogenic region [103].

Similar studies have been reported for identifying specific epitopes of T lymphocytes. For instance, T cell epitopes in CCHFV NP were examined by Goedhals et al. in 2017 in patients who had recovered from the infection. In this regard, the study represents the first documentation of CD8^+^ T cell epitopes in CCHFV NP [96]. The study indicated that T cells primarily respond to NP, highlighting NP as a key structure recognized by most memory T cells. The research also notes that only two regions within Gc trigger T cell responses, whereas no T cell epitopes were observed within Gn. The limited reactivity observed against glycoproteins compared to NP suggests that NP may be a critical antigen for recovery from the disease.

## 7. CCHFV NP Immunology and Diagnostic Tests

Nucleic acid amplification tests and viral antigen detection assays are considered as primary methods used for the diagnosis of CCHF within the first week after the onset of symptoms [104]. As a screening test for acute patient samples, the antigen-capture enzyme immunoassay (EIA), using monoclonal antibodies against recombinant NP, is known to have lower sensitivity compared to nested reverse transcription polymerase chain reaction (RT-PCR) for viral antigen detection [105]. It has also been established that serological tests are more useful for diagnosing CCHF after the first week of the disease [106].

Due to its role as a major protein detectable throughout the viral infection, and its inclusion of immunodominant epitopes that bind to IgG and IgM antibodies against CCHFV, NP has consistently been the preferred viral antigen in the development of many serological diagnostic tests [107,108,109]. Undoubtedly, in the development of tests against CCHFV, preparing antigens without isolating the virus from ticks or infected patients, and without the need for biosecurity concerns or BSL-4 conditions, provides significant advantages. Therefore, recombinant viral NP-based EIAs with high sensitivity and specificity have been developed using expression systems such as bacterial [110,111], plant [112], and insect cells [101,113] to detect CCHFV-specific antibodies in human or animal sera with measurable high efficiency. In a study utilizing bacterial expression to obtain CCHFV proteins, recombinant NP (rNP) and mucin-like variable domain (rMLD)-based EIAs were developed and tested with acute and convalescent patient sera. The results indicated that the combined use of antigens in the developed EIA yielded more sensitive results compared to their individual use. Additionally, anti-CCHFV IgG antibodies were detected with 97% sensitivity in convalescent-phase patient sera, compared to the acute phase. These results indicate that using multiple recombinant protein antigens in CCHFV EIA could provide higher sensitivity [114]. In one study, a test capable of detecting antibodies specific to three recombinant immunogenic CCHFV proteins (NP, GN-ectodomain, and GP38) was reported. This test was described as a suspension microarray-based multiplex test and has been used for screening various types of animal serum samples. The test enabled the simultaneous detection of antibodies against the three proteins within the same sample. The results indicated that there were differences in antibody responses, but that the triplex test was capable of detecting antibodies specific to the three CCHFV proteins [115]. 

The performance of the newly developed CCHFV NP-based EIA tests is usually compared to the existing commercial EIA kits to determine their suitability for seroepidemiological studies [116]. For example, following such an evaluation, the performance of anti-CCHFV IgM and IgG tests for the serodiagnosis of acute CCHFV infection in the Turkish population was reported and described as highly sensitive and comparably specific [117]. Affimers, small proteins that bind to target proteins with affinity in the nanomolar range, were generated for CCHFV NP [118]. The affimers were able to bind to NPs with high affinity, specificity, and were therefore suggested as promising candidates for the development of rapid diagnostic tests through lateral flow assays [118].

## 8. CCHFV NP and Vaccines

The high morbidity and mortality associated with CCHFV infection underscore the urgency of developing effective prevention and treatment strategies against CCHF. In this context, vaccine candidates are being developed using various platforms, and the antiviral potential of certain molecules is being investigated. Due to its potential clinical benefits, the nucleoside analog ribavirin has been included in the WHO’s “Essential Medicines” list [119] as an effective treatment for CCHFV since March 2007. Ribavirin is particularly noted for its use in early treatment and as post-exposure prophylaxis in high-risk situations. However, ribavirin has not been universally accepted, and its efficacy has been questioned [120]. Favipiravir, a pyrazinecarboxamide derivative approved for the treatment of influenza, has been claimed to provide protection against several RNA virus infections, including CCHFV [121,122]. Favipiravir has not yet received widespread clinical approval for human CCHF cases, despite its use in a few cases of COVID-19 co-infected CCHF [123,124]. Recently, an FDA-approved anti-influenza drug, a baloxavir derivative, has been reported to target CCHFV RNA transcription/replication in cell cultures, effectively inhibiting viral infection. This aspect of the study claims to contribute to the design and development of endonuclease-based anti-CCHFV drugs [125]. The inhibition of CCHFV OTU, a viral deubiquitinase, could enhance cellular immune responses against the infection. Studies have shown that the designed ubiquitin variant CC4 is a potent inhibitor capable of blocking CCHFV replication in vitro [126]. One of the major challenges with the therapeutic use of small-protein inhibitors like CC4 is their intracellular delivery. In this context, recent studies have claimed that in vivo treatment with CC4 does not provide protection against lethal CCHFV infection [127].

The importance of developing a vaccine against CCHFV, which poses a threat to three billion people worldwide, is evident [128]. Advancements in biochemical and molecular techniques, along with the development of animal models, have facilitated the evaluation of various CCHFV vaccine platforms in animal models. These include inactivated vaccines, subunit vaccines, VLP–VRP vaccines, viral-vector-based vaccines, plasmid-based DNA vaccines, and RNA/mRNA vaccines [28]. In this review, we focus specifically on vaccine platforms that use CCHFV NP as the antigen. Given the substantial number of studies in the literature where CCHFV NP is used as an antigen in immunization efforts, it is possible that some NP-based studies may have been overlooked in our review. We would like to note that this situation should be considered a limitation and a potential omission of the review. In addition to these studies, recent studies in the literature also includes multiple epitope vaccines (MEVs) against CCHFV developed using reverse vaccinology and immunoinformatics approaches [129,130]. However, it is not yet possible to predict the outcomes of testing these candidates with the virus.

To date, the only CCHFV vaccine applied in humans is an inactivated vaccine derived from newborn mouse brains, used exclusively in Bulgaria [131]. In a study from Türkiye, Canakoğlu et al. reported on a cell culture-based and inactivated Turkey-Kelkit06 CCHFV vaccine using IFNAR^−/−^ mice. The vaccine provided 80% protection following a homologous challenge with a high lethal dose [132]. In a further study comparing the effectiveness of the CCHFV vaccine derived from a mouse brain with the cell culture-based CCHFV vaccine, it was found that the cell-culture-based vaccine generated a more effective CCHFV-specific antibody and T-cell response. The cell-culture-based inactivated CCHFV vaccine demonstrated 100% protective efficacy in immunosuppressed BALB/c mice [133]. 

The CCHFV NP is a viral structure that carries highly conserved B and T cell epitopes with less than 5% amino acid variation between strains. Additionally, NP is a dominant antigen involved in viral replication and transcription, contributing to genome encapsidation and vRNP formation, and stimulating natural, humoral, and cellular immune responses. For all these reasons, CCHFV NP has frequently been used as an antigenic entity in vaccine formulations. Some of these vaccine candidates, utilizing such formulations, have demonstrated significant protection in preclinical studies with animal models (Table 1). Last year, subunit vaccine candidates based on CCHFV NP and CCHFV Gn were developed using the baculovirus–insect expression system (rvAc-Gn, rvAc-NP, rvAc-Gn-NP and Zera-Gn, Zera-NP that contain antigens fused with Zera-peptides). In these studies, it was reported that rvAc-Gn or Zera-Gn induced stronger humoral and cellular immune responses compared to NP responses [134,135]. 

The efficacy and characterization of immune responses induced by a DNA vaccine encoding CCHFV Gn, Gc, and NP in combination with transcriptionally competent virus-like particles (tcVLPs) were investigated in a 2017 study. In the CCHFV challenge experiments in DNA-immunized mice, both antibody responses and predominantly Th1 cell immune responses were induced, providing 100% effective protection. In the same study, despite higher neutralizing antibody levels, only partial protection was observed in tcVLP-immunized mice. The authors concluded that NAbs alone may not be sufficient for protection against CCHFV, and that effective protection may also require Th1 cell responses [92]. In another vaccine study, Hu et al. developed three candidate DNA vaccines encoding CCHFV NP, Gn, and Gc in fusion with lysosome-associated membrane protein 1 (LAMP1). The immune effects and protective elements were evaluated in a transgenic mouse model carrying human MHCs. According to the results, immunizations with LAMP1-CCHFV NP induced both Th1 and Th2 responses, indicating that the DNA vaccine might be an effective candidate to provide complete protection against CCHFV [136]. 

In another DNA vector study, based on the assumption that sufficient antibody responses are not induced against the CCHFV NP if used alone, the CD24 protein, known to have a stimulatory effect on B and T cells, was included in the design with the CCHFV NP. Here, DNA vectors encoding CD24 and NP were prepared separately, and immunizations conducted in two separate regimes, and their cytokine and total/specific antibody responses, were assessed. It was found that NP alone or in combination with CD24 elicited significant cellular and humoral responses in BALB/c mice. It has also been reported that such immunizations provided strong protection in IFNAR^−/−^ mice following a challenge. The protection was characterized by increases in IL-6 and TNF-α levels. Moreover, it was suggested that CCHFV NP-CD24 represents a practical approach for vaccine development against the virus, with CD24 acting as a potential adjuvant producing a markedly higher stimulatory effect [137].

In another vaccine model, cynomolgus macaques were used to test a DNA vaccine containing plasmids encoding CCHFV NP and GPC antigens. The study demonstrated the induction of CCHFV-specific antibody and T-cell responses. Additionally, despite the immunized macaques producing low levels of NAbs against CCHFV, the vaccine was reported to prevent the symptoms of the disease caused by the virus. This study is notable for presenting the first report of an effective vaccine in NHPs [78]. In a secondary study by the same group, the efficacy of DNA vaccines containing CCHFV NP and/or GPC antigens was evaluated in NHPs under two separate regimens. The study observed that a two-dose regimen of NP+GPC combined immunization provided significantly better protection compared to the three-dose immunization with NP or GPC alone. The results were interpreted to indicate that the NP-containing immunogen offers strong protection against CCHFV, with both humoral and cellular immunity contributing to optimal vaccine-induced protection [138]. 

In a study using IFNAR^−/−^ mice with the CCHFV-IbAr10200 strain, it was reported that a single dose (10^5^ TCID_50_ of VRP) of the CCHFV VRP vaccine, carrying viral NP, provided complete protection against the virus [139]. Similarly, in another study, a single dose of VRP immunization was reported to provide heterologous protection against Turkish or Omani strains in IFNAR^−/−^ mice, and that all the animals recovered from the disease [140]. In a separate study evaluating VRP vaccines for rapid immunization protocols, it was reported that a single dose of VRP vaccination spared animals from the clinical disease of a CCHFV challenge at 7 days post-immunization, and against fatal outcomes of CCHFV challenged at 3 days post-immunization [141]. In a recent study, the responses of IFNAR^−/−^ mice immunized with CCHFV NP VRP were compared for immunizations performed 3, 7, 14, or 28 days before challenge. The study reported no significant differences in natural immunity, T cell activation, and anti-NP IgM/IgG antibody titers among these groups. However, the greatest reduction in viral titers was observed in the group where immunizations were conducted 28 days before the challenge. Based on these findings, it has been noted that immunizations administered well before the challenge resulted in the highest anti-NP antibody avidity and complement-mediated effector functions. This may highlight the importance of antibody-mediated responses in providing protection against CCHFV infection using the VRP vaccines [142]. 

As stated earlier, the initial vaccine development studies utilizing DNA virus vectors were reported by Dowall et al. [91]. Despite the robust humoral and cellular responses, protection was not achieved in these studies, where the modified vaccinia Ankara virus was used as a vector for CCHFV NP. In a later study, a new vaccine candidate was designed using another viral vector, human adenovirus 5 (AdV-5), as the carrier and CCHFV IbAr10200 NP as the antigen [143]. In mice immunized with Ad-NP, the development of an anti-NP humoral response was demonstrated. A single dose of the Ad-NP vaccine provided 30% protection against a lethal CCHFV challenge in IFNAR^−/−^ mice. However, this protection could be increased to 78% [143]. In a related study on viral vectors, bovine herpesvirus type-4 (BoHV-4) was tested for developing a vaccine against CCHFV [144]. The BoHV-4 vector vaccine candidate, encoding the full-length NP of CCHFV, provided 100% protection against lethal doses of the CCHFV Ank-2 strain in IFNα/β/γR^−/−^ mice during challenge tests. All these studies have explored viral vector-based vaccine candidates, demonstrating various levels of protection offered by these candidates. 

In another vaccine study model, Sindbis, an RNA virus, was used as vector. In the study, the CCHFV NP was incorporated into a Sindbis vector-based replicon, and its immunogenic potential was investigated [145]. In CCHFV-NP-immunized mice, cytokine production and antibody responses were observed. However, the protective effect of the induced immune responses was not specifically addressed in this study [145]. 

In a similar RNA virus model, an alphavirus-based replicating RNA (repRNA) vaccine platform formed by Venezuelan equine encephalitis virus RNA replicons were utilized and included CCHFV NP (repNP), CCHFV GPC (repGPC), and a combination of both repNP plus repGPC [146]. In the study, a single dose of 100 ng of either repNP or repNP+repGPC antigens provided complete protection against infection in mice. Unexpectedly, the analysis of the immune response elicited by the vaccine components revealed that the immunization stimulated antibody responses against viral NP and cellular immunity against GPC [146]. In a subsequent study by the same group, it was demonstrated that a single dose of the repRNA vaccine provided complete protection through non-neutralizing anti-NP antibodies against viral NP and GPC-specific T cells against GPC, thereby supporting the previous findings. Additionally, a dual-antigen vaccine approach was developed that simplified production by generating two separate RNAs in a single synthesis reaction [147]. In a report where these studies were extended from mouse models to primates, rhesus macaques vaccinated with RNAs expressing CCHFV NP and GPC (repNP+repGPC) demonstrated strong but non-neutralizing humoral immune responses against CCHFV NP. A significant protection against a CCHFV challenge was observed; the authors concluded that repRNA vaccines could provide protection in NHPs [148].

In an mRNA vaccine study, it has been stated that the unmodified mRNA vaccine expressing NP from the non-optimized S segment of the Ank-2 strain of CCHFV provided 50% protection in the single-dose group and 100% protection in the booster group [149]. Similarly, this time, nucleoside-modified mRNAs encoding only CCHFV NP or glycoproteins (GcGn) were administered as antigens within lipid nanoparticles (LNPs) [150]. The study utilized the IFNAR^−/−^ mouse model and the IbAr10200 strain and noted that both vaccine candidates provided full protection against CCHFV. Additionally, mRNA-LNPs induced strong humoral and cellular immune responses in both IFNAR^−/−^ and immunocompetent mice; neutralizing antibodies were not required for protection. In the same study, immunizations with GnGc mRNA-LNP protected mice from the disease and elicited both neutralizing antibodies and T cell responses. However, the NP mRNA-LNP vaccine was stated as the preferred choice due to the immunogenicity of the NP and its highly conserved structure [150]. 

As part of the immunological investigations of NP at our laboratory, an innovative mRNA vaccine candidate has been developed. The design included base modifications and expressed only the CCHFV NP [151]. Immunizations and challenge experiments were conducted using the C57BL/6 mouse model with MAR1-5A3 anti-mouse IFNAR-1 blocking antibodies. Both modified (NP-ΨmRNA) and unmodified mRNA molecules, which expressed the NP of the CCHFV Turkey-Kelkit06 strain, were used in immunizations either in naked form or incorporated into FDA-approved PLGA nanoparticles (NP-ΨmRNA-PLGA). Inactivated CCHFV antigen was also used as a positive control in the immunizations. The effectiveness of the vaccine candidate was compared with non-modified nucleoside mRNAs and other vaccine candidates. The results indicated that both NP-ΨmRNA-PLGA and naked NP-ΨmRNA vaccine candidates afforded robust protection against CCHFV infection. These data further contribute to the existing literature supporting CCHFV NP as a protective antigen.

## 9. Conclusions

CCHFV is a primary pathogen causing yearly epidemics resulting in significant morbidity and mortality in the Middle East, Africa, the Balkans, and Asia, specifically in countries south of the 50° parallel north. The lack of an approved vaccine and effective antiviral treatment necessitates urgent measures to combat the virus. In this review, the immunological role of CCHFV NP is addressed, which is encoded by the smallest gene segment of the virus. Like most viral NPs, it is highly abundant in the virion, participates in genome encapsidation and vRNP formation, and is involved in viral replication and transcription. It interacts with various cellular proteins and factors, playing critical roles in viral pathogenesis. As a dominant antigen, it stimulates innate, humoral, and cellular immune responses. The NP shows less than 5% amino acid variation among CCHFV strains, making it a highly conserved viral element, and carries numerous B and T cell epitopes, demonstrating highly immunogenic properties. Due to these features, the CCHFV NP plays a critical role in the immunology, biology, and pathogenesis of the virus and also serves as a key antigenic target in vaccine formulations.

In this review, the undeniable role of CCHFV NP in the immunology of the virus is discussed in the light of current studies in the literature. According to the available data, antibody, and T cell responses against CCHFV NP appear to constitute the most critical component of antiviral immunity. Therefore, based on a critical evaluation of the literature, it would not be incorrect to suggest that CCHFV NP will be an indispensable element in the vaccines against this virus. While acknowledging the value of intensive research on CCHFV over the past two decades, it should also be noted that there is still much progress to be made in viral immunology, biology, pathogenesis, and efforts to combat this neglected virus.

## Figures and Tables

**Figure 1 viruses-16-01547-f001:**
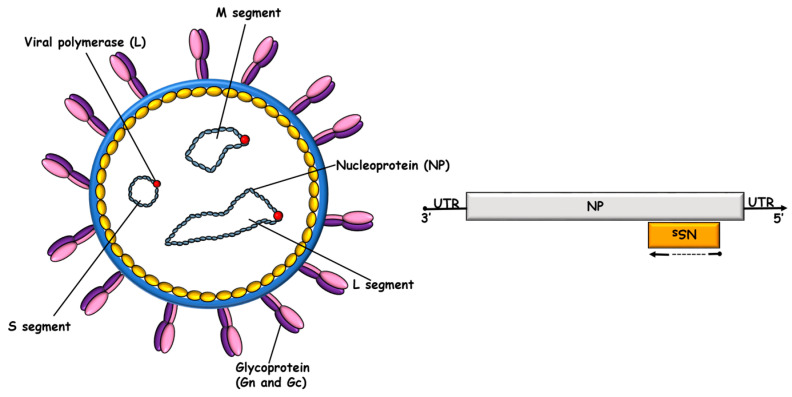
CCHFV virion structure and S segment genome organization. The CCHFV virion contains a tri-segmented, negative-sense, single-stranded RNA (vRNA) genome encapsidated by the nucleoprotein (NP) and the RNA-dependent RNA polymerase (RdRp; L protein). The S segment also codes the non-structural S protein (NSs).

**Figure 2 viruses-16-01547-f002:**
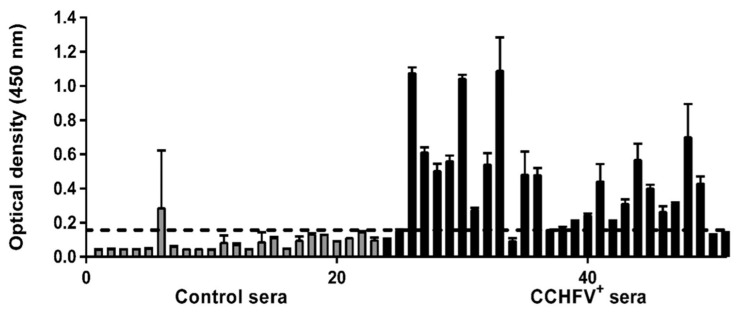
Evaluation of NP-specific antibody response using CCHF convalescent sera. For this purpose, 28 CCHFV-specific antibody-positive and 23 negative human sera were used, and NP-specific IgG was probed. Of the samples tested, the in-house recombinant NP enzyme immunoassay detected a total of 25 out of 28 CCHFV-specific antibody-positive sera as positive and 21 out of 23 CCHFV-specific antibody-negative samples as negative. These results indicated that NP is a strong inducer of the humoral immune response during natural CCHFV infection [79].

**Table 1 viruses-16-01547-t001:** CCHFV NP-based vaccine design platforms.

	Strain Name	Types of Antigen	Animal Model	Vaccination Dose and Regime	Antibody Response	T Cell Response	Challenge	Efficacy, % Survival	Ref.
Inactivated vaccines	Bulgarian V42/81 strain	CCHFV whole antigen Suckling mouse brain-derived	NE	Single or four dose	YES	YES	NE	NE	Mousavi-Jazi et al. (2012) [131]
	Turkey-Kelkit06 strain	CCHFV whole antigen Cell **c**ulture-based (VeroE6)	IFNAR^−/−^ mice	Doses: 5 μg Vaccination: 0, 21, 42 day	YES	YES	Turkey-Kelkit06 strain; 1000 PPFU	60%	Canakoglu et al. (2015) [132]
				Doses: 20 μg or 40 μg Vaccination: 0, 21, 42 day				80%	
			IS-BALB/c mice	Doses: 20 μg Vaccination: 0, 14, 28 day	YES	YES	Turkey-Kelkit06 strain; 100 PPFU	100%	Pavel et al. (2020) [133]
Subunit vaccines	Chinese Xinjiang strain HANM18	NP and Gn Expressed in baculovirus-insect expression system	BALB/c mice	Doses: 10^7^ PFU Vaccination: 0, 14, 28 day	YES	YES	NE	NE	Zhang et al. (2023) [134]
		NP and Gn Fused with Zera tags Expressed in baculovirus-insect expression system		Doses: 10 μg Vaccination: 0, 14, 28 day	YES	YES	NE	NE	Zhang et al. (2023) [135]
DNA vaccines	IbAr10200 strain	NP, Gn and Gc	IFNAR^−/−^ mice	Doses: 50 μg Vaccination: 0, 28, 49 day	YES	YES	IbAr 10200 strain; 400 FFU	100%	Hinkula et al. (2017) [92]
	IbAr10200 strain	NP, Gn and Gc Fused with LAMP1	Human MHC (HLA-A11/DR1) transgenic mice	Doses: 70 μg Vaccination: 0, 3, 6 week	YES	YES	100 TCID_50_ CCHFV tecVLPS	% survival not measured NP provided protection	Hu et al. (2023) [136]
	Ank-2 strain	NP and NP combined with CD24	For immunological response: BALB/c mice For challenge: IFNAR^−/−^ mice	Doses: Totally 50 μg Vaccination: 0, 14 day	YES	YES	Ank-2 strain; 1000 TCID_50_	100%	Farzani et al. (2019) [137]
	Hoti strain	NP and GPC Fused with ubiquitin	Cynomolgus Macaque	Doses: 1 mg each plasmids Vaccination: 3 times, 3 weeks interval	YES	YES	Hoti strain; 100,000 TCID_50_	NP+ GPC provided clinical improvements	Hawman et al. (2021) [78]
		NP and GPC Fused with ubiquitin	Cynomolgus Macaque	Doses: 1 mg both NP plasmid and GPC plasmid Vaccination: 2 times, 3 weeks interval	YES	YES	Hoti strain; 100,000 TCID_50_	Two times vaccination regime provided earlier protection	Hawman et al. (2023) [138]
				Doses: 1 mg NP plasmid or GPC plasmid Vaccination: 3 times, 3 weeks interval					
Virus-like replicon particles (VRP) vaccines	IbAr10200 strain	CCHF tc-VLP components	IFNAR^−/−^mice	Dose: 10^6^ VLPs Vaccination: 0, 28, 49 day	YES	YES	IbAr10200 strain; 400 FFU	40%	Hinkula et al. (2017) [92]
	IbAr10200 strain	CCHF VRP components	IFNAR^−/−^ mice	Doses: 100,000 TCID_50_ Vaccination: Single dose	YES	NE	IbAr 10200 strain; 100 TCID_50_	100%	Scholte et al. (2019) [139]
				Doses: 1000 TCID_50_ Vaccination: Single dose				78%	
				Doses: 100,000 TCID_50_ Vaccination: Single dose	NE	NE	Turkey strain and Oman-97 strain; 100 TCID_50_	100%	Spengler et al. (2019) [140]
							Turkey-2004 strain; 100 TCID_50_	Single dose VRP protected against lethal outcome 3 days post- vaccination	Spengler et al. (2021) [141]
				Doses: 100,000 TCID_50_ Vaccination: 3, 7, 14, or 28 day	YES	YES		Longer vaccination periods provided stronger protection	Sorvillo et al. (2024) [142]
Viral-vector based vaccines	IbAr10200 strain	NP expressed in modified vaccinia virus Ankara (MVA)	IFNα/β/γR^−/−^ mice	Doses:10^7^ PFU Vaccination: 0, 2 week	YES	YES	IbAr 10200 strain; 200 TCID_50_	0%	Dowall et al. (2016) [91]
	IbAr10200 strain	NP expressed in Adenovirus type 5 (AdV5)	IFNAR^−/−^ mice	Doses: 1.25 × 10^7^ IFU; day 0 10^8^ IFU; day 28	YES	NE	IbAr10200 strain; 1000 LD_50_	78%	Zivcec et al. (2018) [143]
	Ank-2 strain	NP expressed in Bovine Herpesvirus Type 4 (BoHV-4)	For immunological response: BALB/c mice For challenge: IFNα/β/γR^−/−^ mice	Doses: 100 TCID_50_ Vaccination: 0, 2 week	YES	YES	Ank-2 strain; 1000 TCID_50_	100%	Farzani et al. (2019) [144]
Self- replicating Alphavirus-based vaccines	SPU 187/90 strain	NP DNA-based Sindbis replicon	NIH-III Heterozygous mice strain	Doses: 100 μg NP Vaccination: 0, 21, 42 day	YES	YES	NE	NE	Tipih et al. (2021) [145]
	Hoti strain	NP and GPCVenezuelan Equine Encephalitis Virus (VEEV) RNA replicon	IS -C57BL/6 mice	Doses: 2.5 μg NP, 2.5 μg GPC, and 5 μg NP + GPC Vaccination: 0, 4 week	YES	NE	UG3010 strain; 100 TCID_50_	100% NP and NP + GPC	Leventhal et al. (2022) [146]
				Doses: 1 μg NP + GPC, two dual RNAs and 0,5 μg repGcFL-NP, bivalent single-RNA Vaccination: Single dose	YES	YES	UG3010 strain; 100 TCID_50_	100%	Leventhal et al. (2024) [147]
			Rhesus Macaque	Doses: 25 μg NP, 25 μg GPC, and 50 μg NP + GPC Vaccination: 2 times, 6 weeks interval	YES	NE	CMP-CCHFV Hoti strain; 100,000 TCID_50_	NP + GPC provided protection	Hawman et al. (2024) [148]
mRNA vaccines	Ank-2 strain	Naked NP mRNA	For immunological response: C57BL/6 mice For challenge: IFNα/β/γR^−/−^ mice	Doses: 25 μg mRNA Vaccination: 0, 2 week	YES	YES	Ank-2 strain; 1000 TCID_50_	Single dose showed 50% protection Double dose showed 100% protection	Farzani et al. (2019) [149]
	IbAr10200 strain	Nucleoside- modified NP and GcGn mRNAs Formulated within LNP	IFNAR^−/−^ mice	Doses: 10 μg each mRNAs and 20 μg mix Vaccination: 0, 3 week	YES	YES	IbAr10200 strain; 400 FFU	100%	Appelberg et al. (2022) [150]
	Turkey-Kelkit06 strain	Nucleoside- modified NP mRNA Formulated within PLGA	For immunological response: BALB/c mice For challenge: IS-C57BL/6 mice	Doses: 10 μg mRNA; day 0 20-25 μg mRNA; day 14 Vaccination: 0, 2 week	YES	YES	Turkey-Kelkit06 strain; 400 PPFU	100%	Keskin et al. (2023) [151]

IS, Immune-suppressed transiently with mAb-5A3 (IFNAR-blocking); NE, Not evaluated; Ref., Reference.

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
