# Peer review of "The Role of Nucleocapsid Protein (NP) in the Immunology of Crimean–Congo Hemorrhagic Fever Virus (CCHFV)"

_viruses, 2024, doi:10.3390/v16101547_

Round 1

Reviewer 1 Report

Comments and Suggestions for Authors

Overall, the paper provides a comprehensive overview of the multifaceted roles of CCHFV NP in viral replication, pathogenesis, and immune responses. However, some aspects of the manuscript could benefit from clarification, expansion, and restructuring to improve its readability and scientific rigor. Below are my specific suggestions for improvement:

As for Introduction

1# Clarify scope and motivation: The introduction briefly mentions the overall structure of the paper, but it could benefit from a clearer statement of the specific research questions this review aims to address and why they are important.

2# Cite relevant background literature: Consider citing more recent and relevant reviews on CCHFV NP to contextualize the work and highlight the gaps your review aims to fill.

As for CCHFV NP in Viral Biology

3# Authors may expand on structural differences: The mention of structural differences between CCHFV NP and other viral NPs is intriguing but brief. Expand on these differences, providing examples and citing specific studies to support your claims.

4# Please clarify RNA-binding mechanisms: The discussion of RNA-binding properties is complex and could be reorganized for clarity. Consider breaking down the mechanisms into separate subsections, with illustrations or diagrams where appropriate.

5# Adding more details on host-virus interactions would benefit: The interactions between CCHFV NP and cellular proteins are of great interest. Provide more details on the specific pathways involved and the potential therapeutic implications.

As for CCHFV NP in Innate Immunity

6# Expand on innate immune receptor involvement: The involvement of innate immune receptors like RLRs, TLRs, and NLRs in CCHFV recognition is briefly mentioned. Discuss these in more depth, including their specific roles in CCHFV infection and how they modulate immune responses.

7# Authors could supplement relevant animal model studies in brief: Support your claims about innate immunity with data from relevant animal models, especially those involving IFN-Alpha/Beta-deficient mice.

8# Discuss cytokine profiles: Further elaborate on the cytokine profiles associated with CCHFV infection and their correlation with disease severity.

For Additional Sections, Formatting, and Clarity

9# Include a section on diagnostic and therapeutic potential: Given the importance of CCHFV NP as a diagnostic target and vaccine candidate, discuss recent advancements and future directions in these areas.

10# Improve figure captions: The figure captions are brief and could be expanded to provide more context and interpretation.

11# Consistency in terminology: Ensure consistent use of terminology throughout the manuscript. For example, use "NP" consistently after first appearance instead of alternating between "NP" and "nucleocapsid protein."

12# Check for grammatical errors and typos: The manuscript contains a few grammatical errors and typos that should be corrected before publication.

13# Summarize key findings and future directions: The conclusion could be strengthened by more explicitly summarizing the key findings of the review and outlining potential avenues for future research.

In summary, the manuscript provides a valuable review of the multifaceted roles of CCHFV NP in viral biology and immunology. With some clarifications, expansions, and restructuring as outlined above, it has the potential to become an impactful contribution to the field.

Comments on the Quality of English Language

There is few spelling mistakes on part-of-speech. Minor editing of English language required.

Reviewer 2 Report

Comments and Suggestions for Authors

The review article by Pirincal and Doymaz entitled “The Role Nucleocapsid Protein (NP) in the Immunology of Crimean-Congo Hemorrhagic Fever Virus (CCHFV)” in viruses focuses on describing the role of NP in various cellular and viral events. The authors have done a very nice job in accumulating the data and presented in a very logical way. But I would like to see the review in a more precise and organized way of telling the NP’s function. The review article is very lengthy about 15 pages and effectively the authors can trim down to 10-12 pages. The table at the end about the vaccine approach is very informative. And I would encourage the authors to either include more tables or schematic representations to summarize what in text.

Major comments

(i)             For the introduction section, it would be good to add more info about the disease caused, geographical distribution, epidemiology in a more interesting way so as to capture the reader’s attention to why we need to read about NP’s. If I’m not mistaken Hyalomma m.marginatum is only one of the subspecies of ticks that causes CCHFV. Mention more about the tick role as well as its geographic distribution, and the potential for CCHFV spread. The last paragraph on various other reviews on different topics of CCHFV is unnecessary and can be removed. You can just mention on what you are going to say in your review.

(ii)           CCHFV NP in Viral Biology: Include a section on replication cycle of the virus, followed by the NP’s role in viral replication cycle (whatever known) and then move to specific roles. Include a schematic representation of life cycle of CCHFV describing NP’s role in each step of replication stages.

(iii)         CCHFV NP in Innate Immunity: Make this section very precise to what is known for CCHFV. Sentences likeHere, only some of the prominent findings regarding CCHFV-innate immunity interactions will be summarized” can be avoided. General information on innate immunity like PRR/PAMPS/ and other things needs to be trimmed down so as to focus on the topic and reduce the length of the review. A schematic representation of what is known would be ideal.

(iv)          CCHFV NP in Humoral Immunity: This section is very nicely written and very precise and I would like to see rest of the sections needs to be described in a succinct way so that the reader doesn’t lose their attention. The sentences starting like “In other study” can be avoided.

Minor comments

(i)             “CCHFV NP” can be written in just short form as NP itself.

(ii)            5’ “CAP” can be in small letters

(iii)          “HAZV” expand

(iv)          English language is fine but needs a mend.

Comments on the Quality of English Language

The article is very informative- needs a thorough trimming to make it shorten and succinct.

Round 2

Reviewer 1 Report

Comments and Suggestions for Authors

The authors addressed most of the comments.